# Androgen Receptor Activation Induces Senescence in Thyroid Cancer Cells

**DOI:** 10.3390/cancers15082198

**Published:** 2023-04-07

**Authors:** Anvita Gupta, Michelle Carnazza, Melanie Jones, Zbigniew Darzynkiewicz, Dorota Halicka, Timmy O’Connell, Hong Zhao, Sina Dadafarin, Edward Shin, Monica D. Schwarcz, Augustine Moscatello, Raj K. Tiwari, Jan Geliebter

**Affiliations:** 1Department of Pathology, Microbiology, and Immunology, New York Medical College, Valhalla, NY 10595, USA; anvita.nymc@gmail.com (A.G.); mcarnazz@student.touro.edu (M.C.); melanie.jones@westpoint.edu (M.J.); z_darzynkiewicz@nymc.edu (Z.D.); dorota_halicka@nymc.edu (D.H.); tjboconnell@gmail.com (T.O.); hong_zhao@nymc.edu (H.Z.); sinadada@uw.edu (S.D.); raj_tiwari@nymc.edu (R.K.T.); 2Department of Medicine, New York Medical College, Valhalla, NY 10595, USA; 3Department of Otolaryngology-Head and Neck Surgery, University of Washington, Seattle, WA 98195, USA; 4Department of Otolaryngology, New York Eye and Ear Infirmary of Mount Sinai, New York, NY 10003, USA; eshin@nyee.edu; 5Department of Medicine, New York University Grossman School of Medicine, New York, NY 10016, USA; mschwarcz@nyulangone.org; 6Department of Otolaryngology, New York Medical College, Valhalla, NY 10595, USA; augustinemoscatello@gmail.com

**Keywords:** thyroid cancer, DHT, androgen receptor, senescence, tumor migration/invasion

## Abstract

**Simple Summary:**

Thyroid cancer is the most common endocrine malignancy and occurs three times more frequently in women than in men. Further, androgen receptor expression is decreased in thyroid cancer cells. These observations suggest that male hormones may play a role in the prevention of thyroid cancer. We show that androgen stimulation of the androgen receptor inhibits the growth of thyroid cancer cells by inducing a state of senescence, in which cells are not killed, but are unable to multiply. Further work will investigate the ability to use androgens to treat thyroid cancer cells, as well as the possibility of treating and eliminating senescent cells using specific drugs.

**Abstract:**

Thyroid cancer (TC) is the most common endocrine malignancy, with an approximately three-fold higher incidence in women. TCGA data indicate that androgen receptor (AR) RNA is significantly downregulated in PTC. In this study, AR-expressing 8505C (anaplastic TC) (84E7) and K1 (papillary TC) cells experienced an 80% decrease in proliferation over 6 days of exposure to physiological levels of 5α-dihydrotestosterone (DHT). In 84E7, continuous AR activation resulted in G1 growth arrest, accompanied by a flattened, vacuolized cell morphology, with enlargement of the cell and the nuclear area, which is indicative of senescence; this was substantiated by an increase in senescence-associated β-galactosidase activity, total RNA and protein content, and reactive oxygen species. Additionally, the expression of tumor suppressor proteins p16, p21, and p27 was significantly increased. A non-inflammatory senescence-associated secretory profile was induced, significantly decreasing inflammatory cytokines and chemokines such as IL-6, IL-8, TNF, RANTES, and MCP-1; this is consistent with the lower incidence of thyroid inflammation and cancer in men. Migration increased six-fold, which is consistent with the clinical observation of increased lymph node metastasis in men. Proteolytic invasion potential was not significantly altered, which is consistent with unchanged MMP/TIMP expression. Our studies provide evidence that the induction of senescence is a novel function of AR activation in thyroid cancer cells, and may underlie the protective role of AR activation in the decreased incidence of TC in men.

## 1. Introduction

Thyroid carcinoma is the most common endocrine malignancy, with an estimated 43,700 new cases and 2120 deaths occurring due to the disease in the USA in 2023 [1]. The incidence of thyroid cancer has increased more than three-fold over the past three decades [2]. Furthermore, the incidence of thyroid cancer in women remains about 3–4-fold that in men, with a delayed peak incidence of about 1–2 decades seen in men [1,3].

Most primary thyroid tumors originate from thyroid follicular cells and develop into epithelial tumors. These cancers develop three main pathological types of carcinoma: papillary thyroid carcinoma (PTC), follicular thyroid carcinoma (FTC), and anaplastic thyroid carcinoma (ATC) [4,5]. PTC constitutes 85–90% of all thyroid cancer cases; it is categorized as a differentiated thyroid cancer (DTC) and shows indolent tumor growth. FTC is also characterized as a DTC, and accounts for 5–10% of thyroid cancer cases. ATC comprises less than 2% of thyroid cancers, but it is completely undifferentiated, aggressive, difficult to manage, and typically arises in the older population, compared to PTC or FTC [6]. The management of these various types of tumor depends on the age of the patients, the tumor size, extra thyroidal invasion, distant metastasis, vascular invasion, and the tumor variant [6,7].

Studies into the molecular pathogenesis of thyroid cancer have revealed that the disease is initiated by genetic alterations and epigenetic perturbations in driver oncogenes or tumor suppressor genes. Mutations in BRAF, rat sarcoma (RAS), and PTEN genes, and translocations in RET (rearranged during transfection)/PTC and paired box 8 (PAX8)/peroxisome proliferator-activated receptor γ (PPARG) are commonly seen to drive tumorigenesis in the thyroid [8,9,10]. Other genes that are altered in thyroid cancers include TP53, IDH1, CTNNB1, and NDUFA13. TP53 encodes the tumor suppressor p53, and its perturbation is observed in 80% of ATC cases [11]. ATC has river mutations similar to PTC, such as BRAF, but also contains additional alterations, including TERT, TP53, NRAS, CDK4, APC, MED12, ERBB2, DIVER1, AR1D1A, and MEN1 [12]. Furthermore, the genomic profiling of ATC indicates a higher genome-wide tumor mutational burden compared to other thyroid cancer subtypes [12]. 

Sex differences have been identified in the oncogenic mutational process of thyroid cancer. Pan-cancer analysis demonstrated that TERT promoter mutations were observed in 64% of male and only 11% of female papillary thyroid cancer samples, and this was associated with increased overall genome-wide mutational burden [13]. 

Sex bias is prevalent in thyroid disorders, with higher prevalence in women in the reproductive age group, with various factors being speculated as reasons for this phenomenon. As ATC occurs in the older population, there is no evident sex bias [6]. However, in PTC, there is general female predominance, with incidence dependent on age, which is suggestive of a hormonal component [14]. Some studies suggest a link between estrogen and increased inflammation and proliferation in human thyroid cancer cells [15,16], while another study reported increased metastasis via estrogen receptor (ER) α and β activation in thyroid cancer cells [17]. The level of ER expression differs in normal thyroid tissue compared with tumor tissue and among different histotypes of thyroid tumors [18]. Well-differentiated thyroid cancers are more often ER-positive and have a higher degree of ER expression compared with undifferentiated or anaplastic cancers [19,20]. 

Limited but conflicting data are available regarding the expression of androgen receptors (ARs) and their role in thyroid cells. Our lab previously demonstrated the presence of ARs both in normal and tumor thyroid tissues, with variable degrees of expression [21]. Papillary and follicular cancer cell lines undergoing testosterone stimulation in vitro were reported to display up-regulation of ARs and proliferation [22]. The same study group also found a varying pattern of testosterone levels and AR status in the thyroid tissues of men and women, quite possibly hinting at the gender-specific incidence of thyroid tumors [23,24]. Studies on DTC tissues showed that ERα positivity, ERβ negativity, and AR expression were associated with a more aggressive phenotype [25].

There is an increasing body of data regarding the anti-proliferative effects of AR stimulation in breast, ovarian, endometrial, and prostate cancers and cell culture models, and the complex role of androgens, including the extensive capacity for crosstalk between steroid receptors [21,26,27,28,29,30,31,32,33,34,35]. We have recently shown that ARs are involved in a regulatory, anti-proliferative pathway in PTC cells, halting cell cycle progression at the G1/S checkpoint [21]. Mirochnik et al. were the first to link the anti-tumor activity of ARs with cellular senescence in vitro and in vivo in AR-transfected PC3 prostate cancer cells [32]. Cellular senescence is described as largely stable, irreversible cell cycle arrest, usually triggered by a multitude of intrinsic and extrinsic stresses, including physical, chemical, and biological stressors [36,37,38]. The role of senescence in halting proliferation and increasing tumor clearance by the immune system via the secretion of inflammatory factors has only recently been investigated, and has picked up scientific momentum, as this knowledge opens new avenues to possible senescence-associated cancer therapies [39]. 

Previous work from our laboratory suggested that androgens and AR activation played an anti-proliferative and regulatory role in a cell culture model of PTC [21]. We now describe further investigations into the mechanism of AR-driven senescence caused by persistent stimulation of the androgen receptor, and its effect on cell mobility and invasiveness, and characterize the senescence-associated secretory profile of AR-induced senescent cells.

## 2. Materials and Methods

### 2.1. In-Silico Methods of Prediction of AR Expression Using TCGA and Wanderer

The web tool Wanderer allows for real time access and visualization of gene profiles obtained from the TCGA Research Network (http://cancergenome.nih.gov/) (accessed on 15 September 2015) [40]. Wanderer was accessed at http://www.maplab.cat/wanderer, accessed on 15 September 2015. The data set of PTC samples and data type ‘Illumina HiSeq RNAseq’ were selected from the dropdown menu of the web tool. The gene name, AR, was then entered as the targeted query. Graphical displays of AR mRNA downregulation in PTC tumors, compared to normal tissue, were obtained. 

### 2.2. Cell Lines and Cell Culture

The ATC thyroid cancer cell line, 8505C, was purchased from DSMZ (Braunschweig, Germany). 8505C contains the BRAF V600E mutation, as well as a C:G-to-G:C transversion at the first base of p53 codon 248 and an allelic deletion of the gene [41]. K1 (GLAG-66) was obtained from Dr. Rebecca Schweppe at the University of Colorado Cancer Center (UCCC). K1 contains the BRAF V600E mutation, the PI3K mutation (Glu542Lys), and the p53 silent mutation (Arg213Arg) [42].

Cells 8505C and K1 were maintained in complete medium defined as RPMI 1640 (Corning Cell Gro. CAT#10-040-CV, Manassas, VA, USA) supplemented with 10% fetal bovine serum (FBS) (Atlanta Biological CAT# S11150, Norcross, GA, USA), 2 mM L-glutamine (Corning Cell Gro, Manasas, VA, USA), and a penicillin (10,000 IU/mL)–streptomycin (10,000 µg/mL) mixture (Corning Cell Gro, Manasas, VA, USA) at 37 °C in 5% CO_2_. The cells were grown to 90% confluence in T-25 (Falcon, Cat #353108, Corning, NY, USA) or T-75 (Thermo Scientific, Cat #156499, Waltham, MA, USA) flasks, and then, passaged.

The cells were cultured by removing the medium and washing once in 2 or 4 mL phosphate buffered saline (PBS without calcium or magnesium; Corning Cell Gro. CAT# 21-040-CV, Manassas, VA, USA). Then, the cells were incubated with 0.5 mL or 2 mL 0.25% trypsin (Corning Cell Gro, Manasas, VA, USA, 25-053-CI) at 37 °C in 5% CO_2_ for 3–5 min. The cells were then disrupted from the surface by tapping the flask. The action of the trypsin was stopped by adding 2 or 4 mL complete medium to the flasks. The cells were centrifuged and resuspended in PBS. For subculturing, the cells were then plated at a dilution of 1:6. Otherwise, the cells were harvested as indicated for each experiment described below. To freeze the cells, they were stored at −80 °C in freezing medium composed of 10% dimethyl sulfoxide (DMSO; Sigma CAT # D5879, St. Louis, MO, USA) and 90% FBS.

We have previously described the generation of AR-transfected cell lines 84E7, K1-lentiAR, and 8505C-lentiAR [21,43]. Briefly, 84E7 cells were derived through the transfection of 8505C cells with pcDNA3.1 containing the androgen receptor, followed by selection and clonal isolation [21]. K1-lentiAR and 8505C-lentiAR were generated via transfection with lentiviral constructs containing the AR, followed by population selection [43].

### 2.3. Treatment Media

For all experiments, the cells were cultured in RPMI 1640 medium without phenol red (Corning Cell Gro, Cat#17-105-CV, Manassas, VA, USA) supplemented with 5% fetal bovine serum unless otherwise noted, 2 mM L-glutamine, and a penicillin (10,000 IU/mL)–streptomycin (10,000 µg/mL) mixture at 37 °C in 5% CO_2_. For the DHT treatment medium, the final concentration of 5α-dihydrotestosterone (DHT; Sigma CAT# A8330-1G, St. Louis, MO, USA) used was 10 nM. Since DHT was dissolved in ethanol (EtOH; Fisher Scientific #64-17-5, Waltham, MA, USA), an equal volume of EtOH was added to the control medium.

### 2.4. Proliferation Assay

Twenty-five thousand to thirty thousand cells were plated in 6-well plates in phenol-free RPMI supplemented with 5% FBS and allowed to adhere overnight. The cells were treated with 0.1% EtOH or 10 nM DHT in phenol-free RPMI supplemented with 5% FBS the following day, and medium was changed every 48 h. The cells were counted every 24 h from 0–8 days using the trypan blue exclusion assay. Following washing once in PBS, the cells were harvested as described above and resuspended in PBS, and trypan blue solution added (Corning, Cat #25-900-CI). The cells that did not uptake up dye were counted as viable cells, while the presence or absence of cells that did take up dye was noted. Counts were averaged and the experiment repeated 2 more times. The experiments were graphed, and growth curves were generated using Microsoft Excel. The growth curves were used to determine the growth rates for the 3 replicated experiments and averaged. 

### 2.5. Cell Morphology Assay and Cellular Size Determination

Twenty-five thousand to thirty thousand cells were plated in 6-well plates in phenol-free RPMI supplemented with 5% FBS and allowed to adhere overnight. The cells were treated with 0.1% EtOH or 10 nM DHT in phenol-free RPMI supplemented with 5% FBS the following day and medium was changed every 48 h. The cells were visualized every 24 h from 0–8 days using a light microscope at 10×–20× to observe the morphological changes. For cellular size determination, cells were visualized at 6 days using a light microscope at 20× magnification, and 5 random measurements of the cell width and length were carried out using the length function of the Axiovision Rel 4.8 program on an Axiovert 200 M microscope (Carl Zeiss Imaging Inc., Thornwood, NY, USA).

### 2.6. Apoptosis Assay Using Immunofluorescent Cytology

Fifteen thousand cells were plated in 6-well plates (Falcon, Cat #353046, Corning, NY, USA) containing sterilized 22 mm × 22 mm coverslips (Fisher Scientific, Cat #12-542-B, Waltham, MA, USA) in phenol-free RPMI supplemented with 5% FBS, and allowed to adhere overnight at 37 °C. The cells were treated with 0.1% EtOH or 10 nM DHT in phenol-free RPMI supplemented with 5% FBS the following day, and the medium was changed every 48 h for 6 days. A separate set of cells were treated with 10 mg/mL cycloheximide for 1 h prior to the assay. They were then removed, and the cells washed with ice-cold PBS. The cells were then incubated with Annexin V-FITC and 20 µg/mL propidium iodide in 1× binding buffer, for 10 min, in the dark, at room temperature, according to the manufacturer’s instructions (ApoDETECTTM Annexin V-FITC kit, Cat #: 33-1200, Invitrogen, Waltham, MA, USA). After incubation, cells were washed once with the 1× binding buffer provided, and 100 µL of 2.5 µg/mL of Hoechst 33,342 solution (Thermo Fisher Scientific, Cat #62249, Waltham, MA, USA) was added to each coverslip to stain the nuclei of live cells, for 10 min, at room temperature, in the dark. After incubation, the Hoechst stain was removed, cells washed once with the 1× binding buffer, and the coverslips were mounted on glass slides (Fisher Scientific, Cat #12-544-3) using SlowFade Diamond Antifade mountant (Thermo Fisher Scientific, Cat #S36963). The coverslips were sealed at the edges using a nail polish, and images were acquired within 5 min of mounting to preserve the integrity of the live cells. Images were taken using the Axiovision Rel 4.8 program under 100× oil immersion magnification on the Axiovert 200 M microscope (Carl Zeiss Micro Imaging Inc., Thornwood, NY, USA). For images, obtained at 20× and 40× magnification, cells were scanned using a Nikon Eclipse Ti inverted fluorescence microscope (Nikon Instruments Inc., Melville, NY, USA), and NIS-Elements Advanced Research software was used. Quantification was performed using Fiji Image J (NIH, Bethesda, MD, USA).

### 2.7. Senescence-Associated Beta-Galactosidase (SAβG) Determination Assay

Twenty-five thousand to thirty thousand cells were plated in six 6-well plates in phenol-free RPMI supplemented with 5% FBS and allowed to adhere overnight. The cells were treated with 0.1% EtOH or 0.1–100 nM DHT in phenol-free RPMI supplemented with 5% FBS the following day, and the medium was changed every 48 h for 0–6 days. Every 24 h, one plate was stained for beta-galactosidase using the senescence β-galactosidase staining kit from Cell Signaling (Cat #9860, Danvers, MA, USA). In brief, wells were washed once with PBS, followed by the addition of 1 mL of fixative solution (2% formaldehyde and 0.2% glutaraldehyde) per well to fix the cells for 10–15 min at room temperature. The wells were then rinsed twice with PBS. One milliliter of beta-galactosidase staining solution (staining solution, solutions A and B, 20 mg/mL X-gal stock solution) was then added per well. The plates were then sealed with parafilm to avoid evaporation and crystallization of the staining solution. The plates were incubated at 37 °C in a dry incubator for 40 h. The staining solution was then removed from the wells, and the wells washed once with PBS. The cells were then overlaid with 70% glycerol for visualization of blue color under a light microscope at 20×, and stored at 4 °C. The number of blue-stained cells were counted from 5 random fields of view in the control and DHT treated wells, and the percentage of senescence was calculated. Bar graphs were generated using Excel to represent the percentage of senescence following various DHT exposure times. 

### 2.8. Laser Scanning Cytometry for Cell and Nuclear Size

Five thousand to ten thousand 84E7 cells were plated overnight in 2-well chamber slides (Thermo Fisher Scientific Cat# 154453, Waltham, MA, USA) in phenol-free RPMI supplemented with 5% FBS at 37 °C. The following day, the cells were treated with 0.1% ETOH or 10 nM DHT in phenol-free RPMI supplemented with 5% FBS, at 37 °C. The cells were harvested on days 2, 5, and 7, by removing the medium, washing once with cold PBS, and adding cold 70% ethanol for 30 min to 2 h, at −20 °C. The chambers were removed from the slide and stored at −20 °C in 70% ethanol. Prior to scanning, the slides were washed once in PBS for 5 min at room temperature. Cell nuclei were stained with 1 µg/mL 4, 6-diamidino-2-phenylindole (DAPI; Molecular Probes, Eugene, OR, USA) for 15 min, and cells were washed once in PBS, followed by mounting using a SlowFade Diamond Antifade mountant (Thermo Fisher Scientific, Cat #S36963, Waltham, MA, USA). Random sections of each slide were scanned, and nuclear fluorescence was measured via LSC (iCysR; CompuCyte, Westwood, MA, USA) utilizing standard filter settings; fluorescence was excited using a violet (405 nm, for DAPI) laser. The intensities of maximum pixels and integrated fluorescence were measured and recorded for each cell in the scanned sections. Enlarged nuclei had a lower DAPI maximum pixels. Histograms were obtained, and cells represented gates based on the nuclear area. The ratio of maximum pixels to nuclear area were calculated for EtOH- and DHT-treated cells and normalized and compared to the ratios of EtOH-treated cells for respective days of laser scanning.

### 2.9. Flow Cytometry Measurements of Total RNA Content, Protein Content, ROS, and Protein Markers of Senescence

Seventy-five thousand 84E7 cells were plated in T-75 flasks in phenol-free RPMI supplemented with 5% FBS and allowed to adhere overnight. The cells were treated with 0.1% EtOH or 10 nM DHT in phenol-free RPMI supplemented with 5% FBS the following day, and the medium was changed every 48 h. On days 1, 3, and 6, cells were harvested as described above. The cells were then resuspended in ice cold PBS for washing and pelleted again. The cells were resuspended in phenol-free RPMI supplemented with 5% FBS for counting, and adjusted to 10^6^ cells/mL. 

For RNA content determination, 0.4 mL acid detergent solution (0.1% Triton-X-100; 0.08 M HCl; and 0.15 M NaCl) [Fisher Scientific Cat # BP151-100, Waltham, MA, USA; Fisher Scientific Cat #A144-500, Waltham, MA, USA; Fisher Scientific Cat # 764-14-5, Waltham, MA, USA] was added to 0.2 mL of cell suspension. Fifteen seconds later, 1.2 mL of acridine orange (AO) staining solution (Sigma-Aldrich, St. Louis, MO, USA)containing 20 µM AO, at pH 6.0, was added. Cellular fluorescence was measured using a FACScan flow cytometer (Becton-Dickinson, San Jose, CA, USA), with excitation at 488 nm and an argon ion laser and measuring green (530 ± 20 nm) and red (>620 nm) fluorescence. Green cell fluorescence is proportional to cellular DNA content, whereas red luminescence correlates with RNA content. Histograms were obtained and gating was performed. The gating strategy was to restrict cells with higher RNA content into separate gates in each treatment type. The values for each replicate were analyzed as a ratio of DHT over EtOH values. Bar graphs were generated using the same method to exhibit changes in RNA content following various DHT exposure times. 

For protein measurement, cells were resuspended in phenol-free RPMI supplemented with 5% FBS for counting, and adjusted to 10^6^ cells/mL in cold 70% ethanol to fix them. We added 1% sulforhodamine B dye (SRB) (Sigma-Aldrich, St. Louis, MO, USA) to the cell suspension and allowed it to incubate for 20 min at room temperature in the dark. After incubation, the cells were washed with ice-cold PBS, and 400 µL of 1 µg/mL 4,6-diamidino-2-phenylindole (DAPI; Molecular Probes, Eugene, OR, USA) solution was added to the cell suspension to stain the DNA. The intensity of cellular fluorescence was measured using a MoFlo XDP (Beckman-Coulter, Brea, CA, USA) high-speed flow cytometer/sorter. DAPI fluorescence was excited using a UV laser (355 nm), and SRB using an argon ion (488 nm) laser. Our analysis of forward light scatter via flow cytometry provides information on cell size, cell cycle, and protein content. All experiments were repeated at least three times, and the representative data are presented. The values for each replicate were analyzed as a ratio of DHT to EtOH values. Bar graphs were generated using the same method to represent changes in total protein content following various DHT exposure times. 

For ROS measurements, the cells were resuspended in phenol-free RPMI supplemented with 5% FBS for counting, and 10^6^ cells were incubated for 60 min with 10 µM 2′,7′-dihydrodichlorofluorescein-diacetate (H2DCF-DA; Molecular Probes, Eugene, OR, USA) dye, which is a cell-permeant non-fluorescent dye, at 37 °C. This dye, upon cleavage of the acetate moiety by the intracellular esterases and oxidation by ROS and peroxides within the cells, is converted to a strongly fluorescent derivative, DCF, and thus, reveals the ROS abundance. 

For protein marker flow cytometry analysis, the cells were resuspended in phenol-free RPMI supplemented with 5% FBS for counting, and adjusted to 10^6^/mL. They were then fixed in 4% paraformaldehyde (Affymetrix, Cat# 19943, Santa Clara, CA, USA) for 15 min on ice, washed once with ice-cold PBS, and resuspended in cold 70% ethanol for 2 h at −20 °C. Fixed cells were washed again with PBS and incubated overnight with primary antibodies diluted 1:100 in 1% BSA solution at 4 °C. The primary antibodies used were p21 (Cell Signaling Technology, Cat# 2947T, Danvers, MA, USA), p27 (Cell Signaling Technology, Cat# 3686S, Danvers, MA, USA), and p16 (Proteintech Group Inc., Cat# 10883-1-AP, Rosemont, IL, USA). Post incubation, the primary antibodies were washed off with PBS, and the cells were incubated with secondary antibodies (1:100) conjugated to Alexa-fluor 488 or 633 (Molecular Probes, Cat# A11008 and A21071, respectively, Eugene, OR, USA), in the dark, for 45 min, at room temperature. Post-incubation, the secondary antibodies were washed off with cold PBS, and 400 µL of DAPI solution was added to the cells and allowed to stain the nuclei for 10 min. The intensity of cellular fluorescence was measured using a MoFlo XDP (Beckman-Coulter, Brea, CA, USA) high-speed flow cytometer/sorter. DAPI fluorescence was excited using the UV laser (355 nm), and SRB using the argon ion (488 nm) laser. Our analysis of forward light scatter via flow cytometry provides information on protein expression and cell cycle. All experiments were repeated at least three times, and the representative data are presented. The values for each replicate were analyzed as a ratio of DHT to EtOH values. Bar graphs were generated using the same method to exhibit changes in senescence markers following various DHT exposure times.

### 2.10. Immunofluorescence Staining

Fifteen thousand cells were plated in 6-well plates containing sterilized 22 mm × 22 mm coverslips (Fisher Scientific, Cat #12-542-B, Waltham, MA, USA) in phenol-free RPMI supplemented with 5% FBS, and allowed to adhere overnight. The following day, the cells were treated with 0.1% EtOH or 10 nM DHT in phenol-free RPMI supplemented with 5% FBS, and medium was changed every 48 h. The cells were treated for 0–6 days. The medium was removed, and the cells washed twice with PBS. The cells were then fixed with ice-cold 4% paraformaldehyde at room temperature for 5–10 min, and then, washed again two times with PBS to remove the fixative. For intracellular proteins, the cells were permeabilized with 0.2% Triton-X (FisherBiotech, Cat# BP151-100, Wembley, Australia) for 10 min at room temperature, and washed once with PBS. The cells were then blocked in 0.1% Triton-X, 10% goat serum, and 1% bovine serum albumin (BSA; Sigma, Cat #A-9418, Lot #77H0702, St. Louis, MO, USA) for 30 min at room temperature. The cells were then incubated for two hours at room temperature or overnight at 4 °C with primary antibody Ars (Cell Signaling Technology (CST), Cat# 3202, Danvers, MA, USA), p21 (CST, Cat# 2947T, Danvers, MA, USA), p27 (CST, Cat# 3686S, Danvers, MA, USA), and p16 (Proteintech Group Inc., Cat# 10883-1-AP, Rosemont, IL, USA), diluted 1:100 or in accordance with manufacturer’s recommended dilution factor, in 1% BSA solution. The cells were also probed for phalloidin (CST, Cat#13054, Danvers, MA, USA), caspase-8 (Bioss Inc., Cat# bs-0052R, Woburn, MA, USA), LC3A/B (CST, Cat# 12741, Danvers, MA, USA), and Ki67 (CST, Cat# 9449, Danvers, MA, USA). The wells were then washed twice with PBS and incubated with the respective secondary antibody conjugated to either Alexa-fluor 488 or 633 (Molecular Probes, Cat# A11008 and A21071, respectively, Eugene, OR, USA), diluted 1:100 in a 1% BSA solution, for 45 min, at room temperature, in the dark. The cells were then washed once with PBS and incubated with 1 µg/mL 4,6-diamidino-2-phenylindole (DAPI; Molecular Probes, Eugene, OR, USA) solution for 10 min, in the dark, at room temperature. The cells were washed once with PBS, and the coverslip was carefully removed from the wells and mounted on premium microscope glass slides (Fisher Scientific, Cat #22-178-277, Waltham, MA, USA) with SlowFade Diamond Antifade mountant (Thermo Fisher Scientific, Cat #S36963, Waltham, MA, USA). Isotypic IgG was used as a control to define background fluorescence. Images were taken using the Axiovision Rel 4.8 program under 100× oil immersion magnification on the Axiovert 200 M microscope (Carl Zeiss Micro Imaging Inc., Thornwood, NY, USA). For images obtained with 20× and 40× magnification, cells were scanned using a Nikon Eclipse Ti fluorescence inverted microscope (Nikon Instruments Inc., Melville, NY, USA), and NIS-Elements Advanced Research software was used. Quantification was performed using Fiji Image J (NIH, Bethesda, MD, USA).

### 2.11. Western Blotting 

84E7 cells were plated at 900,000 cells/T-25 flasks for 1 day of DHT treatment, 300,000 cells/T-75 flask for 3 days of DHT treatment, or 75,000 cells/T-75 flask for 5–6 days of DHT treatment, and were allowed to adhere to the flasks overnight in phenol-free RPMI 1640 supplemented with L-glutamine, penicillin–streptomycin, and 5% FBS. The cells were treated for 1, 3, or 5 days the following day with DHT at a final concentration of 10 nM in medium. Another set of flasks received an identical volume of EtOH as the DHT flasks, serving as a vehicle control. The cells were harvested as described above, washed twice in ice-cold PBS, and pelleted again. The cells were lysed with RIPA Buffer (50 mM Tris-HCl pH 7.4 [Sigma Cat# T3253 St. Louis, MO, USA] 150 mM NaCl, 0.2% sodium deoxycholate [Fisher Scientific Cat# 302-95-4, Waltham, MA, USA], 0.1% SDS [Fisher Scientific Cat# BP166-500, Waltham, MA, USA], 0.5% NP-40 [ThermoFisher Scientific Cat# 85124, Waltham, MA, USA], and 1 μM Pefebloc [Sigma, Cat# 11429868001, St. Louis, MO, USA]) supplemented with protease inhibitors by incubating them on ice for 45–60 min and vortexing them every 10 min. The lysate was passed 5 times through a 23-gauge needle to shear the DNA and, centrifuged at 20,000× *g* for 20 min at 4 °C. The supernatant was transferred to a clean tube, and the protein concentrations were analyzed by adding Bradford Reagent (Bio-Rad, Cat #500-0006, Hercules, CA, USA) to detect protein reads at 595 nm following the Bio-Rad procedure (Bio-Rad, Hercules, CA, USA). Serial dilutions of Bovine Serum Albumin (BSA; Sigma, Cat #A-9418, Lot #77H0702, St. Louis, MO, USA) were used for a standard curve to determine the amount of protein in the samples. A total of 5–10 μg of protein per lane was loaded in all experiments, unless otherwise noted. All SDS-PAGE analyses were performed under reducing conditions (with β-mercaptoethanol) [Sigma Cat# M6250, St. Louis, MO, USA], using 50 V to run the proteins into the stacking gel (4%) and 100 V to run them through the separating gel (10–15%). Protein was transferred from the gel to the PVDF membrane (Millipore, Billeriac, MA, USA) at 220 mA for 2 h at 4 °C. The membranes were cut and incubated in a blocking solution of 5% dry milk reconstituted in TBST (200 mM Tris-HCl pH 7.4, 150 mM NaCl, and 0.05% Tween 20 [Fisher Scientific, Cat# BP337-500, Waltham, MA, USA]) on a shaker for 1 h at room temperature, or overnight at 4 °C (for phosphorylated proteins). The primary monoclonal antibodies used were p21 (CST, Cat# 2947T, Danvers, MA, USA), p27 (CST, Cat# 3686S, Danvers, MA, USA), and alpha tubulin (CST, Cat# 2125S, Danvers, MA, USA). The antibodies were diluted 1:1000 in 3% milk reconstituted in TBST, and incubated with the membrane on a shaker overnight at 4 °C. After incubation with primary antibodies, all membranes were washed 3 times, and then, secondary antibodies were added. All secondary antibodies were goat anti-rabbit HRP-conjugated antibodies (Abcam, Cat# 7171, Cambridge, UK) diluted to 1:5000 or 1:10,000 in 3% milk with TBST and incubated with the membrane for 2 h at room temperature or overnight, at 4 °C, on a shaker. The membranes were then washed three times with 1× TBST, followed by HRP detection, which was performed using a Super Signal Western blot kit (Pierce; Thermo Scientific, Cat# 32109, Waltham, MA, USA). An autoradiography film was exposed to blots for 2 s to 10 min. Densitometry was used to analyze the band intensity using ImageJ software (NIH, Bethesda, MD, USA). All expressions were normalized to beta-tubulin expression, and fold changes were determined relative to the vehicle-treated control for all proteins, except where noted. 

### 2.12. Conditioned Medium Generation for Inflammation Array

Seventy-five thousand 84E7 cells were plated in T-75 flasks in phenol-free RPMI supplemented with 5% FBS and allowed to adhere overnight at 37 °C. The cells were treated with 0.1% EtOH or 10 nM DHT in phenol-free RPMI supplemented with 5% FBS the following day, and the medium was changed every 48 h. The cells were treated for 6 days, and the treatment medium was replaced with EtOH/DHT medium on the 6th day, after washing the cells twice with warmed PBS. The cells were allowed to culture in treatment-free medium for 24 h for conditioned medium generation. Twenty-four hours later, the medium was removed and centrifuged at 4000 rpm for 5 min to remove any cells or debris, and was immediately aliquoted and used or stored at −20 °C.

### 2.13. Human Inflammation Array

The RayBiotech Human Inflammation Array Q1 kit (RayBiotech Inc., Cat #QAH-INF-1-1, Peachtree Corners, GA, USA) was used to quantify the levels of 20 inflammatory chemokines and cytokines in the conditioned medium obtained from 84E7 cells treated for 6 days with 0.1% EtOH, or 10 nM DHT, in phenol-free RPMI supplemented with 5% FBS. The array was performed on each sample as per the manufacturer’s instructions, with a control of conditioned medium generated from 84E7 cultured for 6 days without any treatment. In brief, the glass slide arrays were blocked with blocking solution for 30 min, and 100 μL of conditioned medium/standard cytokines was added to the wells on the slide and allowed to incubate for 2 h at room temperature. The samples were decanted, and the slides washed 5 times for 5 min in wash buffer I, and then, twice with wash buffer II with gentle rocking. This was followed by incubation with primary biotinylated antibody solution for 2 h at room temperature. Washes were repeated as described before, and the slides incubated with Cy3 equivalent dye-conjugated streptavidin antibody solution for 1 h at room temperature. The slides were washed again and dried in a centrifuge. Signals were visualized through the use of a laser scanner equipped with a Cy3 wavelength (green channel) such as Axon 4000B. Data extraction was performed using GAL files that were specific to each array, along with the microarray analysis software found on the RayBiotech website. Q-analyzer software (RayBiotech Inc., Cat #QAH-INF-1-SW, Peachtree Corners, GA, USA) was used to compute the concentrations of chemokines and cytokines in the conditioned medium. 

### 2.14. Human MMP Array

The RayBiotech Human MMP Array Q1 kit (RayBiotech Inc., Cat #QAH-MMP-1-1, Peachtree Corners, GA, USA) was used to quantitate the levels of 7 MMPs and 3 TIMPs in the conditioned medium obtained from 84E7 cells, treated for 6 days with 0.1% EtOH or 10 nM DHT in phenol-free RPMI supplemented with 5% FBS. The array was performed on each sample as per the manufacturer’s instructions, with a control of conditioned medium generated from 84E7 cultured for 6 days without any treatment. In brief, the glass slide arrays were blocked with blocking solution for 30 min, and 100 μL of conditioned medium/standard cytokines was added to the wells on the slide and allowed to incubate for 2 h at room temperature. The samples were decanted, and the slides washed 5 times for 5 min in wash buffer I, and then. twice with wash buffer II with gentle rocking. This was followed by incubation with primary biotinylated antibody solution for 2 h at room temperature. Washes were repeated as described above, and the slides incubated with Cy3 equivalent dye-conjugated streptavidin antibody solution for 1 h at room temperature. The slides were washed again and dried in a centrifuge. The signals were visualized using a laser scanner equipped with a Cy3 wavelength (green channel) such as Axon 4000B. Data extraction was performed using GAL files that were specific to each array, along with the microarray analysis software found on the RayBiotech website. Q-analyzer software (RayBiotech Inc., Cat #QAH-MMP-1-SW, Peachtree Corners, GA, USA) was used to compute the concentrations of the MMPs and TIMPs in the conditioned medium.

### 2.15. Transwell Migration Assay

Corning Biocoat Control Inserts (Corning, Cat #354578, Manassas, VA, USA) with 8 µm pore membrane filters were used for migration assay as per the manufacturer’s protocol. Seventy-five thousand 84E7 cells were plated in T-75 flasks in phenol-free RPMI supplemented with 5% FBS and allowed to adhere overnight at 37 °C. The cells were treated with 0.1% EtOH or 10 nM DHT in phenol-free RPMI supplemented with 5% FBS the following day, and medium was changed every 48 h. On day 6, cells were harvested as described above. The cells were then resuspended in ice-cold PBS for washing and pelleted again. The cells were resuspended in phenol-free RPMI supplemented with 1% FBS for counting, and adjusted to 15,000 cells in 0.5 mL of 1% FBS medium with either 0.1% EtOH or 10 nM DHT. These cells were added to migration chambers, which were rehydrated for 2 h at 37 °C with serum-free, phenol-free RPMI medium, and 750 µL of the growth medium containing 5% FBS was loaded into the bottom chambers. After 12 h of incubation in a humidified tissue culture incubator at 37 °C and 5% CO_2_, the non-migrating cells were removed from the upper surface of the membrane of the chambers by gently scrubbing using a wet cotton-tipped swab. The cells on the lower surface of the membrane were then fixed in methanol (Fisher Scientific, Cat# 67-56-1, Waltham, MA, USA) for 2 min and dried. The cells were then stained in 1% toluidine blue and 1% borax solution for 3 min, followed by washing in distilled water. The inserts were then allowed to air-dry and counted at 10× magnification. The cells in five random fields were counted per chamber and averaged to determine the cells per field of view for each well.

### 2.16. Transwell Invasion Assay

Seventy-five thousand 84E7 cells were plated in T-75 flasks in phenol-free RPMI supplemented with 5% FBS and allowed to adhere overnight at 37 °C. The cells were treated with 0.1% EtOH or 10 nM DHT in phenol-free RPMI supplemented with 5% FBS the following day, and the medium was changed every 48 h. On day 6, the cells were harvested as described above. The cells were then resuspended in ice-cold PBS for washing and pelleted again. The cells were resuspended in phenol-free RPMI supplemented with 1% FBS for counting and adjusted to 15,000 cells in 0.5 mL of 1% FBS medium, with either 0.1% EtOH or 10 nM DHT. Corning Biocoat Invasion Inserts (Corning, Cat #354480, Manassas, VA, USA) were warmed to room temperature for 30 min and used for the invasion assay as per the manufacturer’s protocol. The chambers were rehydrated in serum-free phenol-free RPMI medium at 37 °C for 2 h. The cell suspension was added to the invasion chambers, and 750 µL of the growth medium containing 5% FBS was loaded into the bottom chambers. After 12 h of incubation in a humidified tissue culture incubator at 37 °C and 5% CO_2_, the non-invading cells were removed from the upper surface of the membrane of the chambers by gently scrubbing using a wet cotton-tipped swab. The cells on the lower surface of the membrane were then fixed in methanol for 2 min and dried. The cells were then stained in 1% toluidine blue and 1% borax solution for 3 min, followed by washing in distilled water. The inserts were then allowed to air-dry and counted at 10× magnification. The cells in five random fields were counted per chamber and averaged to determine the cells per field of view for each well. To control for the effect that increased migration has on the invasion assay, the Invasion Migration Index was used to more correctly assess the effect of DHT on invasion potential.

### 2.17. Statistical Analysis

The experiments presented here represent three biological replicates (unless stated otherwise). Statistical significance was determined using a paired Student’s *t*-test in Microsoft Excel, with a probability (‘*p*’ value) ≤ 0.05 leading us to reject the null hypothesis. The asterisks denoting *p*-values signify the following: * *p*-value < 0.05, ** *p*-value < 0.005, *** *p*-value < 0.0005, and **** *p*-value < 0.00001.

## 3. Results

### 3.1. Androgen Receptor Expression Is Decreased in PTC

The higher incidence of TC in women focused our attention on the potentially protective role of androgens and androgen receptor activation in thyroid cancer. We previously showed that AR expression was reduced in 23 PTC samples compared to matched controls [21]. Taking this observation to the broader and more comprehensive TCGA database of PTC samples, AR expression in 498 PTC samples was found to be significantly reduced by approximately 80% (~4.86 log2 reduction; p = 8.3 × 10^−18^), compared to 59 normal thyroid samples in the database. This supports our previous observation, and further suggests a potential role of AR inactivation in thyroid cancer etiology (Figure 1).

### 3.2. DHT Reduces Thyroid Cancer Cell Proliferation through Interaction with the Androgen Receptor

To experimentally determine the effect of androgens on thyroid cancer cells, PTC and ATC cell lines were utilized. While ATC occurs in older individuals in whom sex hormones have a decreased role, the activation of ARs in ATC may possibly serve in a protective or therapeutic capacity [6]. As thyroid cancer cell lines rarely express functional androgen receptors, the anaplastic thyroid cancer cell line, 8505C, was stably transfected with an AR cDNA plasmid, and a clone, 84E7, was used for further analysis [21]. DHT (10 nM) treatment resulted in AR translocation to the nucleus, nuclear enlargement, and reduced proliferation [21]. To confirm and extend these results, a dose–response assessment of DHT-induced growth inhibition of 84E7 was performed. A significant, dose-dependent decrease in androgen-responsive 84E7 cell growth was observed (Figure 2A). As men have free testosterone in the range of 9–37 nM in circulation, and women have a range between 0.5–2.5 nM, all subsequent experiments were performed with 10 nM DHT. This biological level of testosterone in both men and women was non-toxic to cells in culture. In this and other experiments, 10 nM reduced 84E7 proliferation by approximately 80% on day 6 [21]. Parental 8505C cells, without transfected androgen receptors, showed no significant differences in growth rate in the presence or absence of 10 nM DHT (Figure 2B).

To confirm that the DHT-induced reduction in proliferation is due to its interaction with ARs, proliferation studies were carried out in the presence or absence of the AR inhibitor, flutamide. 84E7 cells treated with 10 nM DHT showed a sharp decrease in proliferation, while 84E7 cells grown in the presence of both flutamide and DHT proliferated unchecked, reaching the cell numbers observed in control conditions (flutamide-only-supplemented or EtOH-supplemented medium) (Figure 2C). These data confirm that a DHT-induced decrease in proliferation occurs via interaction with ARs.

To confirm the antiproliferative effect of DHT on plasmid-transfected 84E7 cells, lentiviruses were employed to introduce ARs into K1 and 8505C cells, generating K1-lentiAR and 8505C-lentiAR. Similar to 84E7, treatment with 10 nM DHT reduced 8505C-lentiAR and K1-lentiAR proliferation by approximately 73% and 83%, respectively, by day 6 (Figure 2D,E).

### 3.3. Distinct Morphological Changes Are Associated with Sustained AR Activation

As indicated above, 84E7 cells treated with DHT appeared enlarged, with larger nuclei than EtOH-treated cells. Microscopy indicated that DHT induced a statistically significant increase in cell size (Figure 3A,B). These micrographs exhibited cell enlargement, flattening, vacuolization, and granularization brought about by AR activation over time. The cells appeared pancake-like and were visually greater in size than the EtOH-treated cells. 

Similarly, 8505C-lentiAR and K1-lentiAR cells treated with 10 nM DHT for 6 days were larger than EtOH-treated cells (1.68-fold and 1.57-fold, respectively) (Figure 3C–F). 

### 3.4. DHT Induces Accumulation of Lysosomal Senescence-Associated Beta-Galactosidase (SAβG)

The results thus far indicate that prolonged AR activation significantly reduced proliferation in a concentration-dependent manner, and modulated cell morphology and size. These results are consistent with the induction of senescence. Senescence is accompanied by and measured via the overexpression and accumulation of endogenous lysosomal beta-galactosidase, which is known to accumulate in the lysosomes of senescent cells [44]. Senescence-associated beta-galactosidase accumulated in the lysosomes of DHT-treated 84E7 cells, from 2.47% (EtOH-treated) to approximately 65.5% (DHT-treated; *p* < 0.05), supporting the idea that the cells undergo senescence after a cell cycle halt (Figure 4A,B). This induction of senescence was validated in DHT-treated lentiviral transfected 8505C-lentiAR and K1-lentiAR cells. SAβG staining in 8505C-lentiAR cells increased from 2% in EtOH-treated cells to 93% in DHT-treated cells (Figure 4C,D). SAβG staining in K1-lentiAR cells increased from 1% in EtOH-treated cells to 93% in DHT-treated cells (Figure 4E,F). Thus, the activation of ARs in three tissue culture models, 84E7, K1-lentiAR, and 8505C-lentiAR, resulted in decreased proliferation and the induction of a senescence-like phenotype. To more fully understand the physiologic changes induced by AR activation, 84E7 was chosen for further investigation.

### 3.5. Androgen-Mediated Decrease in Cell Numbers Is Not Due to Apoptosis, Necrosis, or Autophagy

Our results thus far show an 80–90% decrease in the proliferation rate of 84E7 cells when treated with 10 nM DHT over a period of 6 days. Using immunofluorescence, the decrease in proliferation was determined to be accompanied by a decrease in Ki67, a marker of proliferation. (Figure 5A,D). Since a large decrease in 84E7 cell growth was observed to occur due to the long term activation of AR, we wanted to establish that this phenomenon was caused by the stalling of cells at the G1/S checkpoint [21], and not by apoptosis or autophagy. The expression of annexin, caspase-8, and LC3A/B (Figure 5B–D) in DHT-treated 84E7 cells was not significantly different from that in EtOH-treated 84E7 cells, indicating that reduced proliferation was not due to apoptosis, necrosis, or autophagy.

### 3.6. DHT Induces Enlargement of Nuclear Area and Increased RNA and Protein Content in Cells

In addition to the accumulation of lysosomal SAβG, senescent cells have a characteristic “flattened” pancake-like appearance and enlarged, often irregular-shaped nuclei. These features can be efficiently measured using laser scanning cytometry (LSC), which detects the decline in the local density of DNA-associated fluorescence (4,6-diamidino-2-phenylindole (DAPI)) as a function of the intensity of the maximum pixels (amount of DNA per nucleus), paralleled by an increase in nuclear size (nuclear area). The ratio of maximum pixels of DAPI fluorescence per nucleus to the nuclear area is a reliable marker for the degree of senescence. This ratio decreases further as the senescence program intensifies [45]. As microscopy revealed the presence of large nuclei and flattened cells, LSC was used to determine the ratio of DAPI fluorescence per nucleus to the nuclear area, to evaluate the “depth” of senescence occurring following AR activation. Following AR activation of 84E7 cells, the ratio of maximum pixels to the nuclear area decreased as the induction of senescence progressed, and there was an accumulation of cells in the G1 phase of the cell cycle by day 6 (indicated by an accumulation of nuclei with unreplicated DNA) (Figure 6A).

Senescent cells maintain stable diploid DNA content with continuing macromolecular synthesis, even in the absence of mitosis. This leads to unbalanced growth, which is reflected in the increase in cellular and nuclear size, accompanied by increases in total RNA and protein content [46]. As indicated in Figure 6B–E, both RNA and protein content increased in DHT-treated cells, which is concomitant with G1 arrest (demonstrated in the DNA histograms) and consistent with the induction of a senescent phenotype.

### 3.7. DHT Induces Upregulation of Protein Markers of Senescence

Senescence is a complex program initiated by cells in the event of prolonged DNA repair, telomere shortening or nutrient withdrawal. There are various cell cycle checkpoint proteins that are highly expressed in cells undergoing senescence, including p21, p27, and p16, the proteins which are the most critical effectors of G1/S checkpoint cell cycle halt and the senescence program in cells. Previous studies in our laboratory using Western blotting demonstrated that DHT induced a significant reduction in proliferation in 84E7 cells and upregulated p21 and p27, causing a G1/S checkpoint block [21]. To confirm and extend the role of these proteins in DHT-induced senescence, 84E7 cells were analyzed for the expression of p21, p27, and p16 via flow cytometry. Flow cytometry demonstrated that senescence proteins were upregulated (p16, 104%; p21, 57.5%; p27 82.3%) by DHT treatment compared to EtOH treatment, and are commensurate with the morphologic changes in the cells and the reduction in proliferation (Figure 7A–D). To further determine the localization and confirm the increase in these proteins, immunofluorescence microscopy and Western blotting were used. Immunofluorescence microscopy indicated increased expression of these markers, as well as their nuclear localization (Figure 7E). The Western blot results demonstrated increases in p21 (4.89-fold), p27 (1.88-fold), and p16 (2.42-fold) (Figure 7F,G).

### 3.8. AR-Activation Induced Senescent Cells to Promote Paracrine Senescence

Numerous studies have noted that senescent cells can spread growth arrest and senescence to neighboring cells [47,48]. This property is often attributed to the senescence-associated secretory phenotype of the senescent cells, also known as a SASP. Although context-dependent, the SASP has been found to be comprised of inflammatory molecules, soluble and insoluble factors, proteases, and regulators, which induce cell cycle arrest and promote paracrine senescence in the surrounding cells [48]. Conditioned medium from 6-day EtOH- and DHT-treated 84E7 cells, devoid of any EtOH or DHT, was collected and used to culture untreated 84E7 cells for 48 h. The cells were then fixed and stained for SAβG. The presence of SAβG in these cells indicated that the SASP of the senescent cells induced paracrine senescence in conditioned medium-treated cells (Figure 8).

### 3.9. Accumulation of Reactive Oxygen Species (ROS) in DHT-Treated Cells

Intracellular oxidants, such as ROS and peroxides, have been implicated in the induction of senescence [49]. To determine if DHT induced increased levels of ROS during the induction and/or maintenance of senescence, 84E7 cells were treated with 0.1% EtOH or 10 nM DHT, harvested, and incubated with 2′,7′-dihydrodichlorofluorescein-diacetate (H2DCF-DA) dye. The fluorescence intensity of this dye was measured via flow cytometry, and histograms were obtained using the Kaluza overlay technique. As indicated in Figure 9, DHT treatment resulted in the accumulation of high levels of intracellular ROS, which could potentially result in oxidative damage, leading to rapid senescence induction and maintenance.

### 3.10. AR-Induced Senescence in Androgen-Responsive Thyroid Cancer Cells Does Not Incur a Classic Senescence-Associated Secretory Profile (SASP)

A SASP is mainly comprised of inflammatory cytokines and chemokines, such as IL-6, IL-8, CCL-20, CXCL1 (GRO-α), GM-CSF, and MIP, and can result in local and systemic inflammation, disrupt tissue organization, and create a pro-proliferative environment for neighboring malignant cells when persistent [50]. Some of the inflammatory cytokines of the SASP promote epithelial-to-mesenchymal transition, promoting the metastasis of tumor cells [51]. Alternatively, the SASP could affect the neighboring stroma positively by promoting malignant cell clearance by the immune system, if the SASP is localized and time-limited [50]. Components of the SASP, such as IL-6 and IL-8, also reinforce growth arrest and senescence in some senescent cells, thus controlling the tumor volume [50]. To address this context-dependent paradox of SASP activity, we determined the components of the secretory profile of AR-induced senescence in our model, using Quantibody arrays from RayBiotech. The senescence program induced by continuous AR stimulation did not incur a conventional SASP, as few pro-inflammatory cytokines were upregulated (IL-16 and Oncostatin M), and more pro-inflammatory cytokines (IL-6, IL-6Sr, IL-8, TNF, chemokine RANTES, and MCP-1) appeared to be significantly downregulated with DHT treatment (Figure 10). The receptor antagonist of pro-inflammatory cytokine IL-1, IL-1ra, was significantly upregulated, while IL-1a was not detected. Thus, we conclude that prolonged AR-induced senescence does not incur a classical inflammatory SASP with associated tumor-promoting effects.

### 3.11. AR-Induced Senescent Cells Have Altered Migration and Invasion Potentials and Altered MMP and TIMP Profiles

The results above show that AR-induced senescence altered the cytokine profile in 84E7 cells. Conventional SASPs have been known to affect the senescent cell itself and remodel the surrounding tissues. With the observance of a non-inflammatory SASP in our model, we determined the migration and invasion potential of AR-induced senescent thyroid cancer cells. There was a six-fold increase in the migration of cells treated with DHT for 6 days (Figure 11A,B). The percentage invasion, corrected for the DHT-induced changes in migration, was 30.38% for EtOH-treated cells, compared with 26.09% for DHT-treated cells (invasion index = 0.86 (non-significant difference); Figure 11C,D).

Some proteases, such as matrix metalloproteinases (MMPs), are found in SASPs that modulate tumor migration in a coordinated manner with tumor-derived proteases [48,50]. The inhibitors of MMPs, known as tissue inhibitors of metalloproteinases (TIMPs) are usually decreased or absent from SASPs, to mediate maximum migration and invasion capabilities to the tumor cells [48]. Since we observed an alteration in the migratory, but not invasive, properties of cells treated with DHT, we wanted to validate these results by profiling the MMPs and TIMPs in DHT-treated 84E7 cells. MMPs and TIMPs in conditioned medium from 6-day EtOH- and DHT-treated 84E7 cells were quantified using Quantibody arrays. A decline was seen in the levels of MMP-1, MMP-3, MMP-10, and TIMP-4, while a moderate increase was observed in TIMP-1 and TIMP-2 (Figure 12).

## 4. Discussion

Sex disparity has been previously documented in various human malignancies, including thyroid cancer [52]. The American Cancer Society indicates that the female-to-male thyroid cancer incidence ratio is 3–4 to 1 [1]. Despite significant advancements in the illumination of the molecular pathways underlying thyroid cancer development and progression, the influence of sex on the pathological characteristics and outcomes of PTC is largely understudied [53,54,55]. Body weight, body mass index, diabetes, reproductive and menstrual status, environmental and dietary factors, and tumor sex hormone receptor expression are some of the hypotheses that have been investigated for sex differences in PTC initiation and progression; however, limited experimental evidence demonstrates the mechanism responsible for the phenomenon of sex disparity [52]. ATC, seen in older individuals, does not have a sex disparity such as that of PTC, supporting the idea of a hormonal component of the disease [6]. 

While the tumor-modulating roles of sex hormones have been extensively investigated in breast and prostate cancers [56,57], their roles in the regulation of gene expression and cancer pathology are now emerging in non-endocrine-related cancers, such as colorectal, liver, bladder, and head and neck cancers, and endocrine-related cancers, such as endometrial and ovarian tumors [58,59,60,61,62,63]. The effects of estrogen, mediated by its alpha and beta receptors, have long been found to play a proliferative role in thyroid and breast cancers [64]. Studies of papillary thyroid tumors have suggested a dramatic increase in cellular proliferation, migration, and invasion in thyroid cancer cell lines treated with estrogen, in comparison with treatment with testosterone [20,65,66]. 

An androgen receptor (AR) is a nuclear receptor found in the cytoplasm, and is an androgen-activated transcription factor that translocates into the nucleus upon binding to testosterone or dihydrotestosterone. It regulates gene expression programs that are particularly important for the male phenotype. Like other steroidal pathways, AR signaling does not function in isolation, but rather, in conjunction with multiple signaling pathways. The proliferative effects of androgens are well-documented in various malignancies. Karla Kohan-Ivani et al. reported that androgens may induce a direct decrease in the levels of p21 protein, driving proliferation in epithelial ovarian cancer [67]; meanwhile, Pietri et al. present a rationale for using AR antagonists for AR-targeting treatment as a new therapy for triple-negative breast cancer, where AR signaling drives tumor progression [68,69].

AR modulates cellular proliferation and metastasis in androgen-dependent and -independent prostate cancer cells, and hence, is a major drug target to manage the disease [70]. An investigation of AR gene expression’s impact on clinical features and the progression of PTC demonstrated an association with high cancer risk and extrathyroidal extension in PTC [71]. Although there have been some reports of the anti-proliferative effects of androgens in prostate cancer cells and mesenchymal cells [72,73], research into this role of ARs in other cancers has been very limited. 

In our study, we present data that indicate that ARs exhibit decreased expression in PTC. Further, we describe an anti-proliferative facet of the androgen receptor in the thyroid cancer cell lines 8505C and K1, which we engineered to express a functional AR gene to render it androgen-responsive (84E7, 8505C-lentiAR, and K1-lentiAR). Prolonged activation of the receptor, achieved through the addition of 10 nM dihydrotestosterone in culture medium over a period of up to 6 days, resulted in significant reductions in the cellular numbers of all three cell lines. This effect was determined to be DHT- concentration-dependent, AR-specific, and was accompanied by a reduction in proliferation potential, as observed by the decrease in Ki67 in 84E7 cells. Autophagy and apoptosis were not the causes of decreased proliferation. 84E7 cells treated with DHT also showed a distinct cellular morphology when compared to the control (EtOH) cells. They were significantly larger, highly granular, and vacuolarized, suggesting a senescence-mediated process. Along with the loss of proliferation potential, the AR activation of all three AR-expressing cell lines induced senescent cells, which exhibited an accumulation of beta-galactosidase in the cytoplasm. Significant increases in nuclear area, RNA and protein content, and reactive oxygen species were observed in 84E7. An upregulation of senescence protein markers p16, p21, and p27 was also observed in 84E7 cells. Thus, one mechanism by which androgen/AR signaling causes senescence is prolonged cell cycle arrest, which the cells are unable to escape from. The association of the androgen/AR complex with the senescence program was first made in androgen-responsive prostate cancer cells [32,74] and human dermal papilla cells [75]. However, there were few reports that demonstrated the association of AR antagonists [76] and androgen depletion in prostate cancer [77] with senescence. Our study is the first to demonstrate the induction of a senescence program in androgen-responsive thyroid cancer cells with prolonged activation of ARs by DHT. The possibility of the induction of AR expression and activation as a therapeutic approach may be an exciting avenue for study.

Senescence is generally accompanied by inflammatory cytokine production by the senescent cells, which signal the host immune cells to eliminate damaged cells. Upon profiling the senescent cell secretome of our cells, we found that AR-induced senescent cells secreted significantly lower levels of inflammatory cytokines than control cells, suggesting that AR-dependent senescence incurs an anti-inflammatory SASP, which reduces tumor-promoting chemokines. Conditioned medium derived from senescent cells induced paracrine senescence in naïve 84E7 cells, indicating the potential of SASPs to induce a senescent phenotype in neighboring, untreated cells. These data also suggest that AR activation in thyroid cancer cells favors a more anti-inflammatory environment, allowing us to understand why men have a lower incidence of inflammatory diseases and cancers of the thyroid gland compared to women [1].

Inflammation is a well-established risk factor for many malignancies, and there are sex differences in thyroid inflammation that begin around puberty [78]. Thyroid inflammation due to autoimmune disorders is more common in women, and several studies have demonstrated that women with these disorders have an increased risk of thyroid cancer [79,80]. With that, investigations into the role of sex hormones in inflammation and the role of AR activation have uncovered mechanisms, including the inhibition of NFKB activation, and the suppression of T cells, macrophages, neutrophils, and natural killer cells, and hence, a reduction in inflammatory cytokine production [81,82,83,84]. 

Interestingly, migration was found to be upregulated six-fold with continuous AR activation in 84E7 cells, but no significant change in the invasion potential of these cells was observed. Our data indicate that the lack of increased invasive potential of DHT-treated cells can be attributed to a decline in the levels of MMP-3, MMP-10, and TIMP-4, while a moderate increase is observed in MMP-1, TIMP-1, and TIMP-2. The data also account for the observation of the high lymph node metastasis of thyroid cancer in men, as cells with high migratory and low invasive properties can easily gain access to the lymph nodes; this is because of the presence of a discontinuous, leaky basement membrane of the lymphatic system [85]. 

The concept of pro-senescence therapy has been gaining momentum recently as a novel therapeutic approach to control tumor cells. It is now becoming more evident that cellular senescence is one of the primary and most potent physiological anti-neoplastic responses initiated by the cells to counteract oncogenic drivers, and can be cleared, in vivo, via a robust innate immune response [86]. Senescence has been shown to affect numerous pathways in various cancers, including p53, MAPK, NFKB, and mTOR, consequently inhibiting proliferation and altering inflammatory processes [87,88,89]. While the AR induction of cellular senescence can be interpreted as a tumor-suppressive phenomenon, the clearance of senescent cells may be desirable, as these cells have the potential to survive indefinitely. In clinical settings, the presence of chemotherapy-induced senescent (CIS) cells is correlated with chronic inflammation, incomplete tumor clearance, and disease relapse [51]. The selective elimination of senescent cells has become imperative and challenging, as they do not show susceptibility to non-specific genotoxic chemotherapies. Studies are now focusing on therapeutically exploiting the hypercatabolic nature of CIS with synthetic lethal metabolic targeting, or inhibiting lysosomal ATPases [45]. It is imperative to determine if DHT-induced senescence has clinical implications in tumor treatment therapy and whether these cells could be selectively eliminated in vitro by senolytic drugs that inhibit lysosomal ATPases. There are three main strategies for the utilization of senolytic drugs, including (1) selectively killing pro-tumorigenic senescent cells, (2) promoting the secretion of beneficial SASP factors, and (3) inhibiting the secretion of pro-tumorigenic SASP factors [88,89]. Future directions include investigating the senolytic effects of macrolides on our AR-induced senescent cells. Senescent cells could also be targeted for senoptosis by inhibiting survival pathways and anti-apoptotic mechanisms.

Engaging immune cells and promoting their function, while also engaging pro-senescence therapy and responses, may prove to be beneficial for the clearance of senescent cells, as well as rapid reductions in tumor burden. Adoptive cell transfer therapies involving the administration of activated and expanded tumor-reactive T cells may prove to be a useful strategy to promote immune cell function in patients undergoing pro-senescence therapies. In co-culture systems and in vivo, senescent cells could be targeted for clearance by T cell targeting, NK cells, antibodies, and antibody-mediated drug delivery [50]. Thus, a combination of immune-modulatory agents with pro-senescence agents, apoptosis inducers, and SASP modulators should be considered when evaluating pro-senescence modalities as a tool for effective cancer management. Further, we have observed that AR activation decreases the expression of PDL-1 in 84E7 and K1-lentiAR cells, rendering cells more susceptible to immune destruction by T cells [43].

The effective and optimized use of senescence for therapy would require the sensitive detection and quantification of senescence in vivo, and the recruitment of effector immune cells to the tumor microenvironment. IHC analysis on tissue biopsy samples may be performed to detect and quantify SAβG, the upregulation of markers such as p16, p21, and p27. In vivo imaging of SaβG-positive tumors using the novel fluorescent galactosidase conjugate DDAOG (7-hydroxy-9H-(1,3-cichloro-9,9-dimethylacridin-2-one galactosidase) can be utilized [86]. The measurement of the expression of ARs in tumor tissue and serum levels of circulating active testosterone could potentiate the development of personalized hormonal therapy for the management of thyroid cancer in patients. In clinical settings, based on AR expression status in the thyroid tumor, a pro-senescence approach could be taken in combination with traditional treatments of radioiodine ablation, TSH suppression, thyroid hormone replacement therapy, and BRAF inhibitors, in order to reduce tumor mass before lobectomy or thyroidectomy. This may be particularly useful for patients with metastatic PTC, radioiodine-refractory thyroid cancer, or undifferentiated ATC. In the initial stages, the induction of AR-driven senescence in tumor cells may reduce tumor growth itself, followed by the cumulative effect of promoting cells towards apoptosis en route to senescence due to the combination of senescence therapy with radiotherapeutic and chemotherapeutic protocols.

A desirable feature of pro-senescent therapies against cancer is the possibility of a subsequent complementary treatment to completely eliminate the senescent cells [39]. In this regard, it has been found that senescent cells sustain high levels of proteo-toxicity, and require high lysosomal activity [45]. Senescent cells are particularly sensitive to chemical inhibitors of lysosomal ATPases, and this property serves as an Achilles heel in senescent cells [45]. Understanding the crosstalk between senescence and regeneration will provide us with better knowledge to design effective pro-senescent or anti-senescent therapies [45].

The current understanding of the genomic and non-genomic activities of ARs is incomplete. The lack of standardized AR measurement methods and the small number of patients tested for AR expression limits our knowledge of AR expression in various disease states. One way to detect AR activity would be to detect the expression of androgen-responsive genes, such as FKBP5 or RHOB (which we found in our in vitro model system to be responsive to AR-activation) [21]. One could also supplement this information by detecting circulating free androgen in patients and drawing a direct correlation between the levels of active androgen and the expression of AR gene readout proteins. Several studies have documented the association between AR splice variants and prostate cancer progression, including a group that published findings of AR splice variants binding to constitutively open chromatin to promote abiraterone resistance in prostate cancer [90]. While there is a paucity of such studies on thyroid cancer, it would be worthwhile to investigate any AR splice variant that leads the tumorigenesis of PTC and ATC.

Contributing to these limitations, there are contrasting reports documenting increased and decreased expression of ARs in thyroid cancer. Our study, and the field, would benefit from determining AR protein expression in a large cohort of PTC and ATC patients using the IHC of TC patient samples and normal, matched thyroid tissue, with stratification performed based on TNM staging, BRAFV600E mutation, and inflammation status.

It would be useful to validate the increased transcript levels of inflammatory molecules at the protein level. Further, a comparison of our data with an independent RNA/protein database of TC patients would support their potential role in the etiology of TC. Bisulfite sequencing may be performed to detect and validate the methylation of ARs in the thyroid cancer cell lines that have silenced androgen receptors. While androgen-dependent prostate cancer is managed via resection and androgen manipulation, there is a scarcity of studies on testosterone manipulation in other cancer models [91]. The importance of senescence has been underappreciated, and the use of senescence induction for therapeutics remains underexploited. We investigated the convergence of AR signaling with senescence pathways in a thyroid cancer model. Cellular metabolism, mitochondrial function, mitophagy, and miRNA are known to be altered in senescent cells. Exploration into the modulation of crosstalk between androgen/AR complex and these signaling pathways through androgen manipulation in thyroid cancer merits future research. Furthermore, expanding this work into animal models for an in vivo validation of the anti-proliferative and pro-senescent role of ARs would provide extensive depth of knowledge on the role of this nuclear receptor. Although there is still a lot to learn from in vitro and in vivo modeling of senescence, the use of AR-targeted pro-senescence therapy with conventional cancer therapies might minimize toxicity and enhance clinical outcomes and quality of life for cancer patients.

It is tempting to speculate that the manipulation of androgens and androgen receptors may be utilized for therapeutic intervention for thyroid cancer. Further, the use of senolytic drugs, in conjunction with androgen-induced senescence, is an extension of our data. Future directions include investigating the effect of senolytic drugs on androgens/androgen receptor-induced senescent cells.

## 5. Conclusions

We have addressed the observed sex disparity in the incidence of papillary thyroid cancer and demonstrated that the activation of androgen receptors results in the induction of senescence in three cell line models. This was evidenced by G1 growth arrest with significantly increased p16, p21, and p27 expression, accompanied by a flattened, vacuolized cell morphology, with enlargement of the cell and nuclear area, as well as an increase in senescence-associated β-galactosidase activity, total RNA and protein content, and reactive oxygen species. A non-inflammatory senescence-associated secretory profile was induced, significantly decreasing inflammatory cytokines and chemokines, and reducing the propensity for inflammation. Our studies provide evidence that the induction of senescence is a novel function of AR activation in thyroid cancer cells and may underlie the protective role of AR activation in the decreased incidence of PTC cancer in men.

## Figures and Tables

**Figure 1 cancers-15-02198-f001:**
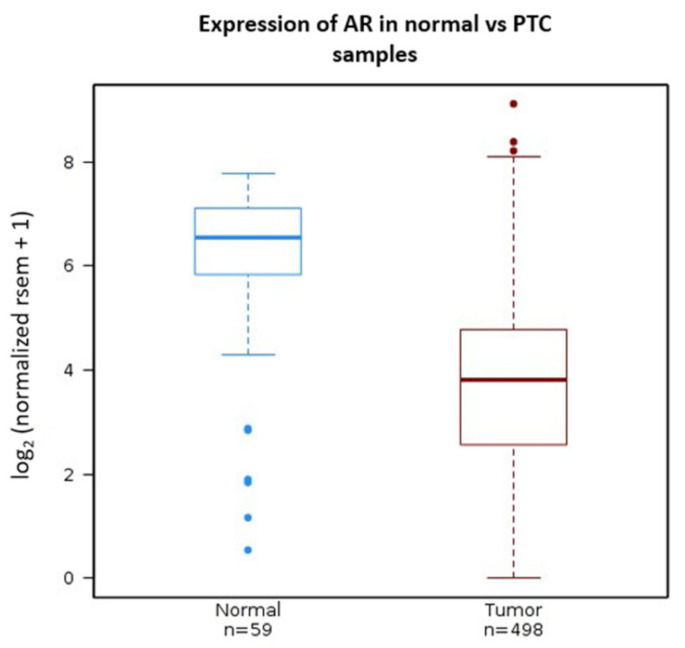
Expression of ARs in PTC samples compared to normal control tissue: TCGA data visualized for AR RNA in tumor and normal samples using Wanderer, a Maplab tool for TCGA RNA data visualization, demonstrated a statistically significant decrease in AR expression in PTC (Wilcoxon p = 8.323325 × 10^−18^).

**Figure 2 cancers-15-02198-f002:**
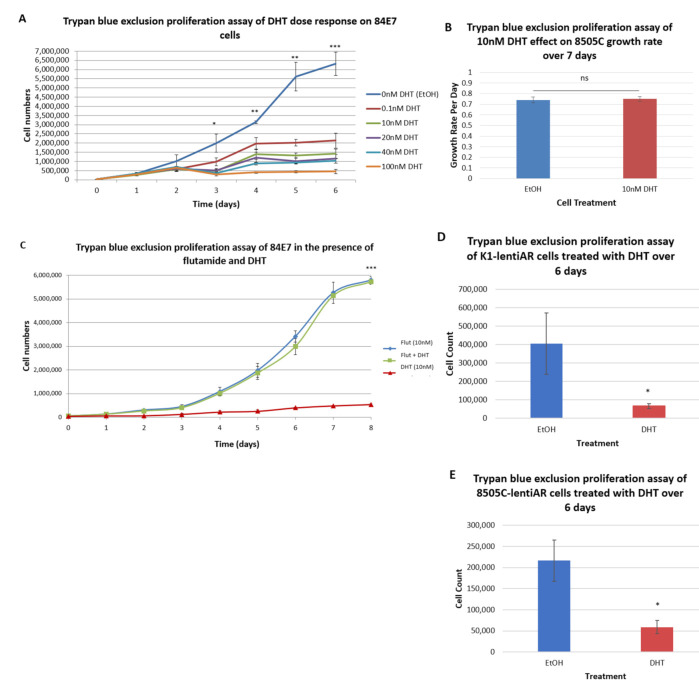
Effect of AR activation on cell proliferation: (**A**) Prolonged AR activation reduced androgen-responsive 84E7 cell growth in a concentration-dependent manner. (**B**) Parental androgen-unresponsive 8505C cells continued to proliferate at the same rate upon prolonged AR activation. (**C**) In the presence of both flutamide and DHT, 84E7 cells continued to proliferate and reach confluency. (**D**) AR activation reduced proliferation in lentivirus-induced androgen-responsive thyroid cancer cell line, 8505C-lentiAR. (**E**) AR activation reduced proliferation in lentivirus-induced androgen-responsive thyroid cancer cell line, K1-lentiAR. Data are expressed as raw numbers obtained from cell counts, and the asterisks denote statistically significant differences (* *p*-value < 0.05, ** *p*-value < 0.005, *** *p*-value < 0.0005).

**Figure 3 cancers-15-02198-f003:**
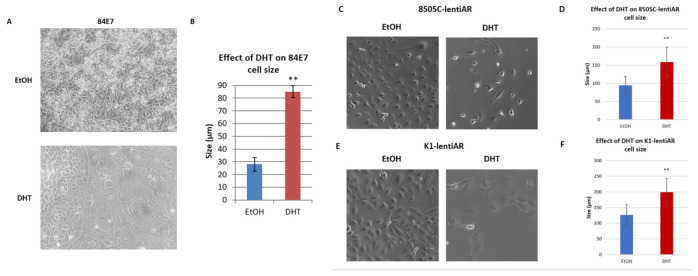
Effect of prolonged androgen receptor activation on cell morphology and size: (**A**) 84E7 cells enlarged in size by day 9 of continuous androgen receptor activation. (**B**) Quantification of 84E7 AR-activation modulation of cell size. (**C**) 8505C-lentiAR cell morphology and size were modulated by day 6 of continuous androgen receptor activation. (**D**) Quantification of 8505C-lentiAR cell size with 6 days of AR activation. (**E**) K1-lentiAR cell morphology and size were modulated by day 6 of continuous androgen receptor activation. (**F**) Quantification of K1-lentiAR cell size with 6 days of AR activation. Raw numbers are representative of unbiased cell dimension measurement in five random fields of view per sample. The asterisks indicate a statistically significant difference (** *p* < 0.005) seen between the control and treated cells. Results represent 3 separate experiments performed on different days at 20× magnification.

**Figure 4 cancers-15-02198-f004:**
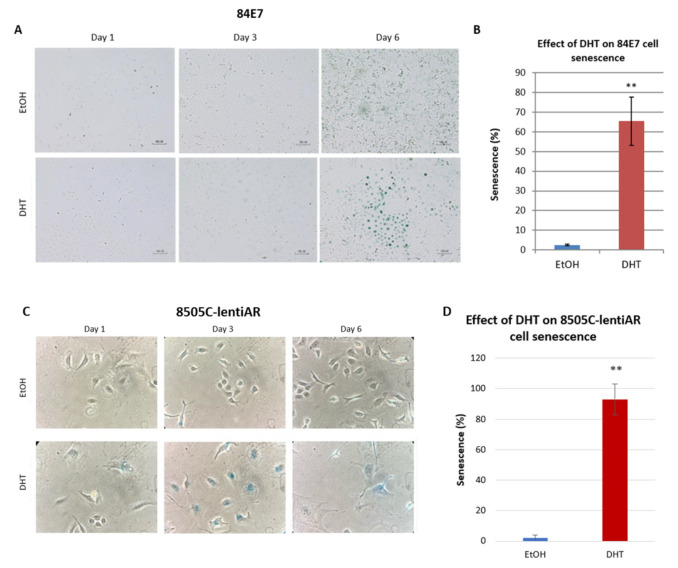
Effect of AR activation on SAβG accumulation: (**A**) AR activation of 84E7 drove significant SAβG accumulation. Top panel shows EtOH-treated 84E7 cells, which exhibited non-specific blue staining for beta-gal. Bottom panel shows blue staining of lysosomes in 84E7 cells treated with 10 nM DHT. (**B**) Quantification of increased senescence-associated beta-galactosidase expression in AR-activated 84E7 cells. (**C**) AR activation drove significant SAβG accumulation in 8505C-lentiAR when compared to EtOH-treated control cells. (**D**) Quantification of increased senescence-associated beta-galactosidase expression in AR-activated androgen-responsive 8505C-lentiAR cells. (**E**) AR activation drove significant SAβG accumulation in K1-lentiAR when compared to EtOH-treated control cells. (**F**) Quantification of increased senescence-associated beta-galactosidase expression in AR-activated androgen-responsive K1-lentiAR cells. The asterisks denote a statistically significant difference between the two groups (** *p* < 0.005). Images were taken at 20× magnification (**A**) and 40× magnification (**C**,**E**).

**Figure 5 cancers-15-02198-f005:**
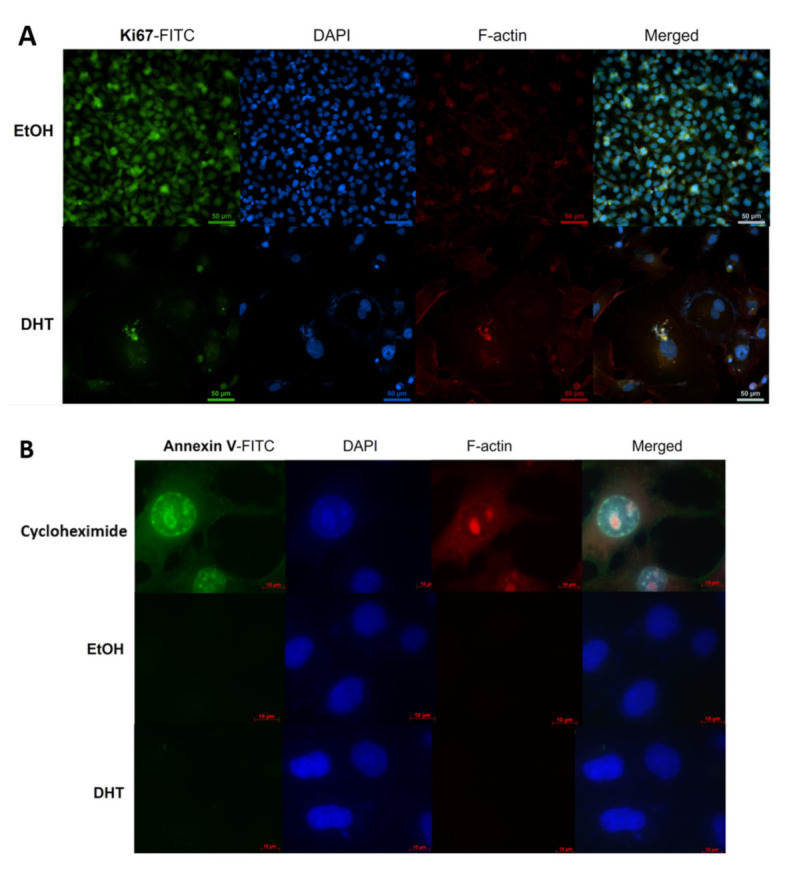
Effect of 84E7 AR activation on markers of proliferation, apoptosis, necrosis, and autophagy. (**A**) AR activation reduced proliferative potential, demonstrated by decreased expression of Ki67, a marker of proliferation. (**B**) AR activation did not cause apoptosis, demonstrated by lack of positive staining using Annexin-V or PI. Cells treated with cycloheximide as a positive control showed green staining fluorescence for Annexin-V (apoptosis), and red staining for PI (necrosis) (top panel). Nuclear DNA was stained blue by Hoechst 33,342 stain. (**C**) Continuous AR activation did not initiate apoptosis or autophagy in 84E7 cells, indicated by a lack of significant change in expression of caspase-8 and LC3A/B. (**D**) Quantification of immunofluorescence data revealed a significant decrease (36%; * *p* < 0.05) in proliferation marker Ki67 and no significant change in apoptosis, necrosis, or autophagy markers.

**Figure 6 cancers-15-02198-f006:**
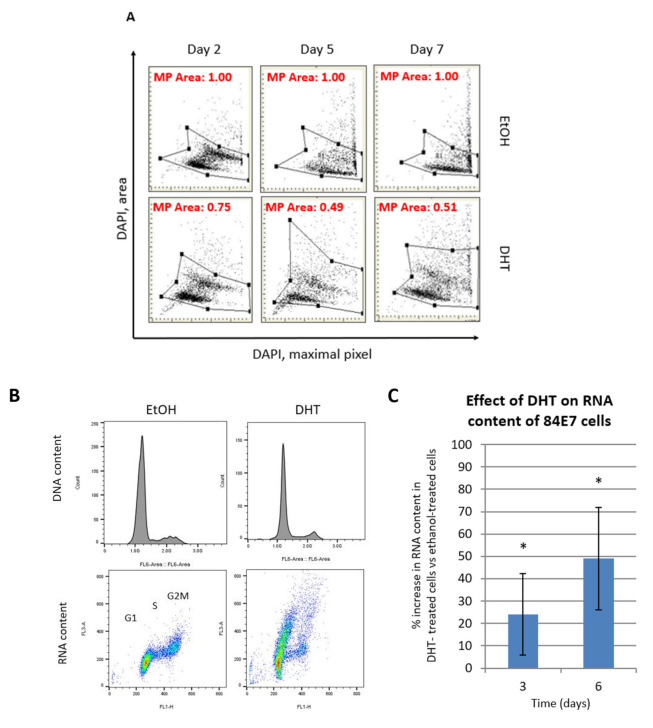
Effect of AR activation on 84E7 nuclear area, DNA, RNA, and protein content. (**A**) Analysis of nuclear changes, measured via laser scanning cytometry, revealed that the ratio of maximum pixels (MP) of DAPI fluorescence per nucleus to the nuclear area decreased, and indicated that the nuclear area increased as the ARs were activated for longer periods of time. (**B**) Flow cytometry demonstrated an increase in total RNA content in 84E7 cells treated with DHT on day 6, as measured via acridine orange staining. DNA histograms demonstrate G1 arrest typical for senescence. (**C**) Quantification of the flow cytometry data revealed a significant 23% and 49% increase in RNA in DHT-treated 84E7 cells over EtOH-treated controls on days 3 and 6. (**D**) Flow cytometry demonstrated an increase in total protein content in 84E7 cells treated with DHT on days 3 and 6, as measured via sulforhodamine B staining. (**E**) Quantification of the flow cytometry data revealed a significant 14.67% and 19% increase in cellular protein in DHT-treated 84E7 cells over control on day 3 and day 6, respectively (* *p*-value < 0.05, ** *p*-value < 0.005).

**Figure 7 cancers-15-02198-f007:**
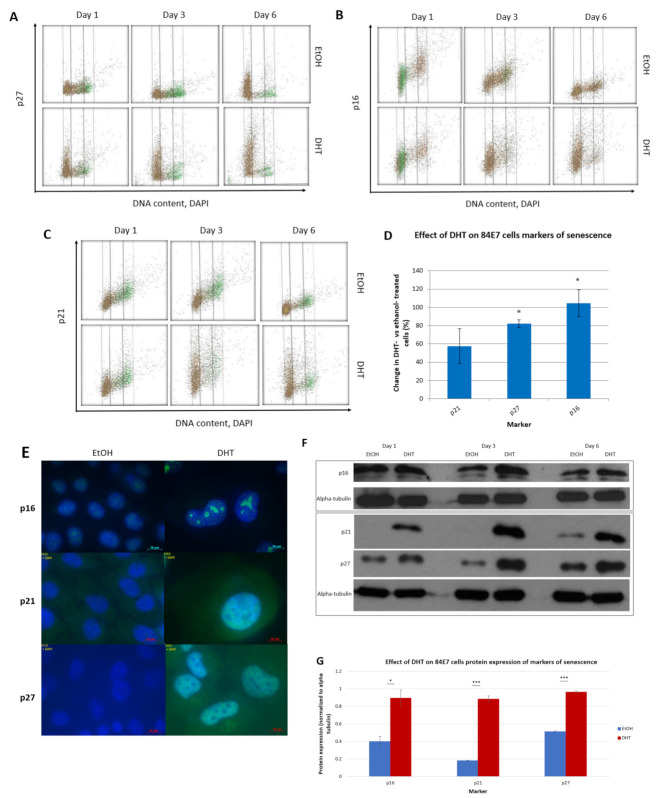
Effect of 84E7 AR activation on protein markers of senescence: (**A**–**C**) Flow cytometry data show the increased expression of senescence markers p21, p27, and p16 in 84E7 cells treated with DHT for 6 days. (**D**) Quantification of accumulation of the senescence proteins. (**E**) Localization of markers p21, p27, and p16 (merged images) in 84E7 cells treated with EtOH (left panel) or DHT (right panel) for 6 days. (**F**,**G**) Western blots and accompanied bar graph demonstrate the upregulation of markers for senescence by day 6 of AR activation with 10 nM DHT. Alpha-tubulin was used as a loading control. Asterisks indicate a statistically significant difference (* *p* < 0.05, *** *p* < 0.0005) seen between EtOH- and DHT-treated cells.

**Figure 8 cancers-15-02198-f008:**
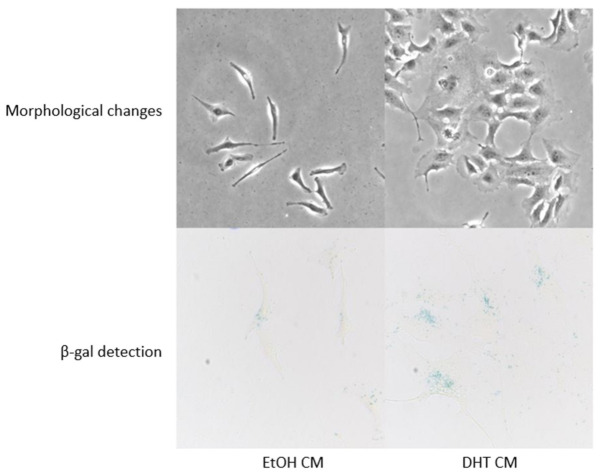
Effect of senescent 84E7 cells’ conditioned medium on untreated 84E7 cells. Conditioned medium obtained from 84E7 cells treated with 10 nM DHT for 6 days induced paracrine senescence in untreated 84E7 cells. Images were taken at 40× magnification.

**Figure 9 cancers-15-02198-f009:**
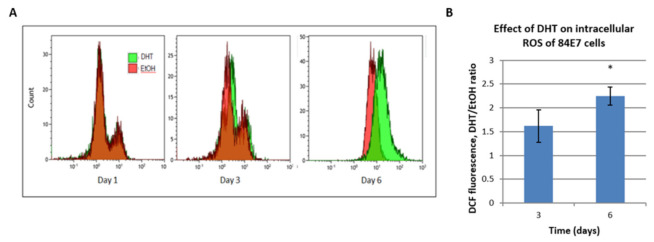
Effect of continuous AR activation on the induction of reactive oxygen species (ROS) in 84E7 cells: (**A**) Flow cytometry demonstrated that AR activation progressively increased ROS content in 84E7 cells, as measured using H2DCF-DA dye. Note the *Y*-axis scale difference. (**B**) Quantification of the flow cytometry data revealed a significant > 2-fold increase in intracellular ROS levels in DHT-treated 84E7 cells over control on day 6 (* *p*-value < 0.05).

**Figure 10 cancers-15-02198-f010:**
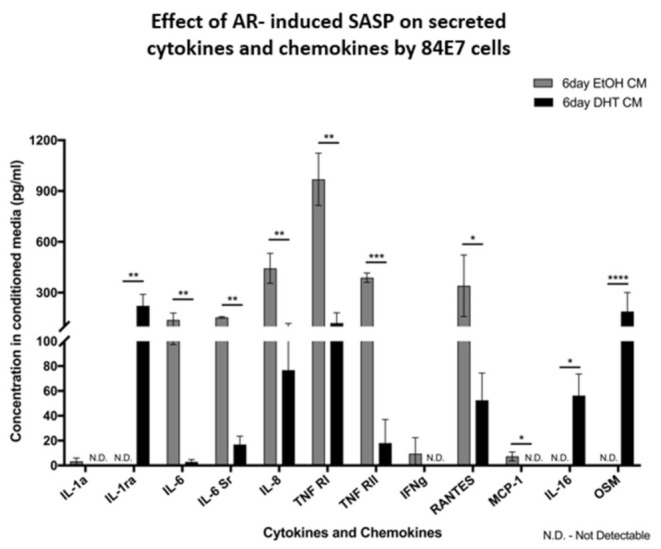
Senescence-associated secretory profile (SASP) of AR-activated 84E7 cells. Quantification of cytokines and chemokines in the conditioned medium showed significant suppression of inflammatory cytokines IL-6, IL-6Sr, IL-8, TNF, chemokine RANTES, and MCP-1, and an increase in anti-inflammatory cytokine IL-1ra, revealing a non-inflammatory SASP (* *p*-value < 0.05, ** *p*-value <0.005, *** *p*-value < 0.0005, **** *p*-value < 0.00001).

**Figure 11 cancers-15-02198-f011:**
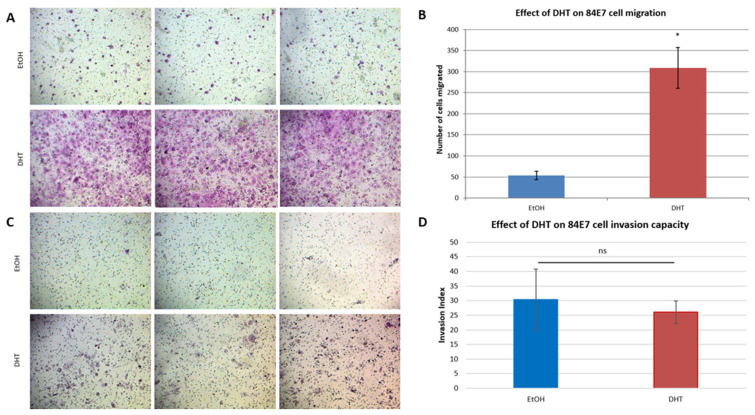
Migratory and invasive potential of 84E7 cells with AR activation: (**A**) Representative images of migrated 84E7 cells in the presence and absence of AR activation at 10× magnification. (**B**) Quantification of migrated cells. A six-fold increase in migration was observed in DHT-treated cells. (**C**) Representative images of 84E7 cells invading through the lower surface of the Matrigel insert membrane at 10× magnification. (**D**) Invasion index of EtOH- and DHT-treated 84E7 cells. Invasion index is calculated as mean number of cells invading the Matrigel insert membrane divided by the mean number of cells migrating through the control insert membrane ×100. EtOH-treated = 30.38; DHT-treated = 26.09. No significant change in the invasion potential of AR-induced senescent cells (* *p*-value < 0.05, ns = not significant).

**Figure 12 cancers-15-02198-f012:**
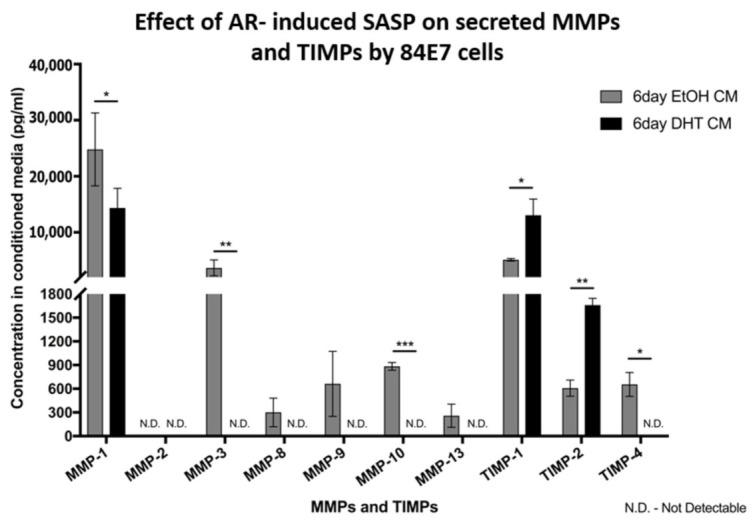
Effect of AR-activation on MMP and TIMP levels in the SASP. Continuous activation of ARs in androgen-responsive 84E7 cells shows a significant decrease in MMP-1, -3, and -10 and a moderate increase in inhibitors TIMP-1 and TIMP-2 (* *p*-value < 0.05, ** *p*-value < 0.005, *** *p*-value < 0.0005).

## Data Availability

Data can be accessed from corresponding author upon request.

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
