# Peer review of "Androgen Receptor Activation Induces Senescence in Thyroid Cancer Cells"

_cancers, 2023, doi:10.3390/cancers15082198_

Round 1

Reviewer 1 Report

This study show that androgen stimulation of the androgen receptor inhibits the growth of thyroid cancer cells by inducing a state of senescence, and indicated that induction of senescence is a novel function of AR activation in thyroid cancer cells and may underlie the protective role of AR activation in the decreased incidence of PTC cancer in men.

1. As the author says, 8505C is an anaplastic thyroid cancer cell line, it is unreasonable to explain the sex difference of PTC.

2. Keep the introduction brief

3. Discuss more about critical functions in senescence in cancer and how they can interact with other molecular pathways. Check all sections of in terms of spell and grammar mistakes.

Author Response

We would like to thank you for the additional time to properly complete the revision of our manuscript.   We would also like to thank the reviewers for their thoughtful suggestions, which have been incorporated int the revised manuscript.  The suggestions has added greatly to the quality of the manuscript.

Below is an item-by-item list of responses/revisions.

REVIEWER 1:

  1. As the author says, 8505C is an anaplastic thyroid cancer cell line, it is unreasonable to explain the sex difference of PTC.

  1. This has been corrected in the Introduction, Results and Discussion sections of the manuscript. Sex (and hormone) differences occur in PTC, and ATC, seen in older individuals, does not. This supports the idea of a hormonal component of the disease. 

  1. Keep the introduction brief.

  1. We have reduced the content of the introduction, and, as suggested by another reviewer, have moved some content to the discussion section.

  1. Discuss more about critical functions in senescence in cancer and how they can interact with other molecular pathways.

  1. We have added this material to the Discussion section.

  1. Check all sections of in terms of spelling and grammar mistakes.

  1. The manuscript has been edited for Spelling, Grammar, and Syntax.

Again, we extend our thanks and appreciation to the Editors and Reviewers for their patience and very useful and positive suggestions that have been incorporated into the revised manuscript.

Respectfully,

Jan Geliebter, PhD

Professor

Basic Sciences Building 311

[email protected]

Reviewer 2 Report

I read with interest this manuscript on thyroid cancer and the link between the activation of androgene receptor and senescence. The current investigation comes from a laboratory with an excellent expertise in this specific field. Overall, there are significant novelties in this research that may merit publication after some revisions, as suggested:

-fig.1 and fig. 6 should be supplementary figures;

-please expand the discussion regarding sex differences in terms of prognosis of cancer also in general / also discussing different mutational signatures based on sex

-please expand the part on the discussion on regarding inflammation and AR activation that appears associated with and anti-inflammatory environment in thyroid cancer cells

-please discuss in depth the potential therapeutic implications derived from the study of senescence in thyroid cancer

Author Response

We would like to thank you for the additional time to properly complete the revision of our manuscript.   We would also like to thank the reviewers for their thoughtful suggestions, which have been incorporated int the revised manuscript.  The suggestions has added greatly to the quality of the manuscript.

Below is an item-by-item list of responses/revisions.

REVIEWER 2:

The manuscript has been edited for English and Grammar and Syntax.

  1. 1 and fig. 6 should be supplementary figures.

  1. We have retained Figure 1 as it extends our observation to much larger, publicly available database, and significantly supports the basis of our research
  2. We agree that Figure 6 was not optimal as a “standalone.” We have merged it with the material previously in figure 7 and created a new figure 6 with demonstrating the effect of AR activation on 84E7 nuclear area,  DNA, RNA and protein content

  1. Please expand the discussion regarding sex differences in terms of prognosis of cancer also in general / also discussing different mutational signatures based on sex

  1. This has been added to the Introduction.

  1. Please expand the part on the discussion on regarding inflammation and AR activation that appears associated with and anti-inflammatory environment in thyroid cancer cells.

  1. This has been added to the Discussion.

  1. Please discuss in depth the potential therapeutic implications derived from the study of senescence in thyroid cancer.

  1. This has been added to the Discussion.

Again, we extend our thanks and appreciation to the Editors and Reviewers for their patience and very useful and positive suggestions that have been incorporated into the revised manuscript.

Respectfully,

Jan Geliebter, PhD

Professor

Basic Sciences Building 311

[email protected]

Reviewer 3 Report

Dear Authors,

Nice, interesting. logically and carefully elaborated manuscript, thank you! I really liked it very much!

However I have some small remarks, what should be fruitfiled to improve the quality even more:

1) you have nice micrographs of the cells, thank you. BUT, - please indicate magnification (X) or scale bar (up to you) for the following Figures: Figs. 3 A, C, E; Figs. 4 A, C, E; Figs. 5 A, B, C; Fig. 8A; also Fig. 9 (here, please, also correct the Legend to the illustration as it is not enough to mention "...induced paracrine senescence in untreated...", but you have to describe the visible morphological changes into the micrographs; also Figs. 12 A, C;

2) Discussion is amazing, I enjoyed reading of it very much, thanks! Just I would like to ask you to add Limitation paragraph at the end of it, if there are some Limitations what to say about (almost all is done, I think, but perhaps you can say some words here...)! And the other thing, - please, remove the second paragraph from the Conclusions also for the end paragraph of Discussion. These speculations better will fit there!

3) Conclusions, please, shorten, make more precise, remove result description, for instance, the first sentence.

4) Finally, - also References section is excellent. I just would like to ask about 3 previous century sources, - are you sure that you really need them? Perhaps there is such a possibility to remove or exchange them with the most modern ones, as they do not fit a little bit for otherwise very nice manuscript. I do nit insist in this point, but think about it, OK!

Merry Christmas and have  a nice New Year with producing of similar excellent manuscripts into the field!

Author Response

We would like to thank you for the additional time to properly complete the revision of our manuscript.   We would also like to thank the reviewers for their thoughtful suggestions, which have been incorporated int the revised manuscript.  The suggestions has added greatly to the quality of the manuscript.

REVIEWER 3:

The manuscript has been edited for English, Grammar, and Syntax.

  1. Please indicate magnification (X) or scale bar (up to you) for the following Figures: Figs. 3 A, C, E; Figs. 4 A, C, E; Figs. 5 A, B, C; Fig. 8A; also Fig. 9 (here, please, also correct the Legend to the illustration as it is not enough to mention "...induced paracrine senescence in untreated...", but you have to describe the visible morphological changes into the micrographs; also Figs. 12 A, C;

  1. These corrections have been made. Some of the Figures have been renumbered.

  1. Add Limitation paragraph at the end.

  1. Limitation are now clearly discussed at the end of the Discussion.

  1. Remove the second paragraph from the Conclusions also for the end paragraph of Discussion. These speculations better will fit there!

  1. The discussion section has been reorganized to contain Discussion, Limitations and Conclusions.

  1. Conclusions, please, shorten, make more precise, remove result description, for instance, the first sentence.

  1. This has been done as part of the reorganization of the Discussion.

  1. References section is excellent. I just would like to ask about 3 previous century sources, - are you sure that you really need them? Perhaps there is such a possibility to remove or exchange them with the most modern ones, as they do not fit a little bit for otherwise very nice manuscript.

  1. This has been done.

Again, we extend our thanks and appreciation to the Editors and Reviewers for their patience and very useful and positive suggestions that have been incorporated into the revised manuscript.

Respectfully,

Jan Geliebter, PhD

Professor

Basic Sciences Building 311

[email protected]